# A Deep-Learning-Based Approach for Aircraft Engine Defect Detection

**Anurag Upadhyay, Jun Li \*, Steve King and Sri Addepalli** 

School of Aerospace, Transport and Manufacturing (SATM), Cranfield University,
Cranfield MK43 0AL, Bedfordshire, UK
\* Correspondence: jun.li@cranfield.ac.uk

**Abstract:** Borescope inspection is a labour-intensive process used to find defects in aircraft engines that contain areas not visible during a general visual inspection. The outcome of the process largely depends on the judgment of the maintenance professionals who perform it. This research develops a novel deep learning framework for automated borescope inspection. In the framework, a customised U-Net architecture is developed to detect the defects on high-pressure compressor blades. Since motion blur is introduced in some images while the blades are rotated during the inspection, a hybrid motion deblurring method for image sharpening and denoising is applied to remove the effect based on classic computer vision techniques in combination with a customised GAN model. The framework also addresses the data imbalance, small size of the defects and data availability issues in part by testing different loss functions and generating synthetic images using a customised generative adversarial net (GAN) model, respectively. The results obtained from the implementation of the deep learning framework achieve precisions and recalls of over 90%. The hybrid model for motion deblurring results in a $10\times$ improvement in image quality. However, the framework only achieves modest success with particular loss functions for very small sizes of defects. The future study will focus on very small defects detection and extend the deep learning framework to general borescope inspection.

**Keywords:** borescope inspection; images; motion deblurring; U-Net; GAN

## 1. Introduction

Regular maintenance is required for an aircraft engine to stay in a state of continued airworthiness. The work often includes visual inspection, borescope inspection, non-destructive functional testing, etc. Borescope inspection is used to detect defects in hot sections of aircraft engines that are difficult to inspect without disassembling. The current practice for borescope inspection is to insert a flexible camera into the inspection port on an engine. Images are captured by the camera and displayed on a portable monitor, then manually inspected by technicians for signs of defects. It is a labour-intensive process. The results often depend on the judgment of the technicians performing the process. There are cases that the inspection errors leave defects on aircraft engines unnoticed. With advances in data analytics and deep learning, there is a demand in the industry to transform manual aircraft maintenance into an automated, effective and predictive process.

Automated aircraft engine defect detection methods are commonly classified into non-destructive evaluation (NDE), classic computer vision and deep learning methods [1]. The traditional NDE methods [2–4], now with a resurgence in smart sensing [5–7], have achieved good performance in the internal defect detection of aircraft engine blades. However, they are not suitable for blade surface defect detection, due to the size, shape and other characteristics of the defects. While there have been studies [8–11] using classic computer vision techniques (e.g., bilateral filtering and edge detection) to automatically detect defects in aircraft components, the methods face challenges such as manual feature engineering

and lack of model adaptability. Although deep learning methods have been developed, there is very little research applied to borescope inspection. Recently, Wong et al. [12] and Li and Wang et al. [13] proposed methods based on Mask RCNN and customised YOLOv5s, respectively, showing the promise of deep learning techniques for automated defect detection.

This research aims to automate aircraft engine defect detection for borescope inspection by combining state-of-the-art deep learning techniques and classic computer vision methods based on real borescope inspection images. As defects are detected by an impartially trained deep learning model, potential human errors and bias are eliminated. The remainder of this paper is organised as follows: Section 2 provides an overview of the existing work in the area; Section 3 defines the deep learning framework with components such as data acquisition, image processing based on computer vision, image segmentation using a customised U-Net [14] and synthetic image generation using GAN; Section 4 presents the model results; Section 5 discusses the development of the methods; Section 6 concludes the work.

## 2. Literature Review

Borescope inspection, which allows in situ aircraft engine inspection, is one of the most commonly used techniques in aircraft maintenance [15]. Researchers are developing various methods to automate the borescope inspection process using both classic computer vision and cutting-edge deep learning techniques [8,9,15–17].

Aust et al. [8] developed a method to identify defect edges in high-pressure compressor blades with small datasets. The contours of the blades are detected using classic computer vision techniques, including bilateral filtering, edge detection and adaptive thresholding. The feature points containing defects are then calculated and clustered using the DBSCAN clustering algorithm. However, the method is only designed to detect edge defects as a type of error, and it is difficult to apply it to other types of defects. Shao et al. [9] developed a method for inspection image processing based on erosion and histogram equalisation. Instead of using traditional Canny edge detection, which applies Gaussian smoothing and dual thresholding to detect and join edges, the method adopts a Gaussian filter with adaptive adjustment and maximum between-class difference thereby preserving the edge details. Li [10] developed the aero-engine fault diagnosis expert system by integrating the image analysis with the expert system diagnosis based on automatic extraction of the damage feature points and measurement of the internal damage cracks of the aero-engine blades. Ma et al. [18] employed the edge pixel information to identify potentially problematic blades and an HOG descriptor to distinguish the defective and normal blades. Tian et al. [19] used vibrothermography images to construct a dynamic threshold using the signal-to-noise ratio, thus allowing images with defects to be filtered.

These heuristic algorithms developed using computer vision come with various challenges. For example, the defects come in different shapes and sizes, thus manually generating features may become difficult as the data size increases. In addition, some of the defects are small compared to the images, which are difficult to detect based on computer vision. Deep learning models embed automatic feature selection and account for some hidden features that manual feature engineering may overlook. Shang et al. [16] developed a deep learning method based on Mask RCNN to find multi-type defects in borescope inspection. Li and Li et al. [17] developed a similar approach using coarse and fine representation to segment defects from background images. Although the techniques work well on the simulated dataset, they are not accurate for datasets with different data distributions. Li and Wang et al. [13] used a customised YOLOv5s model to detect defects with multiple shapes to obtain an mAP50 accuracy of 83.8%. However, the researchers used a self-built small dataset and only considers five types of defects of aero-engines. Recently, researchers are developing hybrid methods by combining traditional computer vision and deep learning methodologies. For example, Kim et al. [15] integrated the principal component analysis

(PCA) method and scale-invariant feature transform (SIFT) for feature extraction and image pre-processing with a CNN model to categorise the images into defects and non-defects.

This review and analysis of the various research and techniques on defect detection highlights the following issues. Firstly, there is a lack of a comprehensive framework specifically dealing with the borescope images for defect inspection. Secondly, the low resolution and motion blur in images are not explicitly addressed. Thirdly, the computer vision and deep learning methods suffer from the small size of training samples; therefore, synthetic images are required. This research aims to develop a deep learning framework for aircraft engine defect inspection using real borescope inspection images. The framework also resolves low-resolution images, processes motion blur and generates synthetic images.

## 3. Methodology

This research develops a novel framework or a machine learning pipeline that describes and guides the process of aircraft engine defect detection through implementing and orchestrating multiple machine learning components, as shown in Figure 1. The images are firstly collected, pre-processed and fed into the deep learning model for defect segmentation using a bespoke U-Net [14], whose results are then analyzed and measured using predefined metrics such as model accuracy and robustness. Where further pre-processing is required, the images undergo a deblurring and sharpening process, in which the images are deblurred using a combination of computer vision methods and a GAN model given the presence of motion blur, and image edges are enhanced by a sharpening filter. The process is repeated until the desired defect segmentation accuracy and model robustness are achieved.

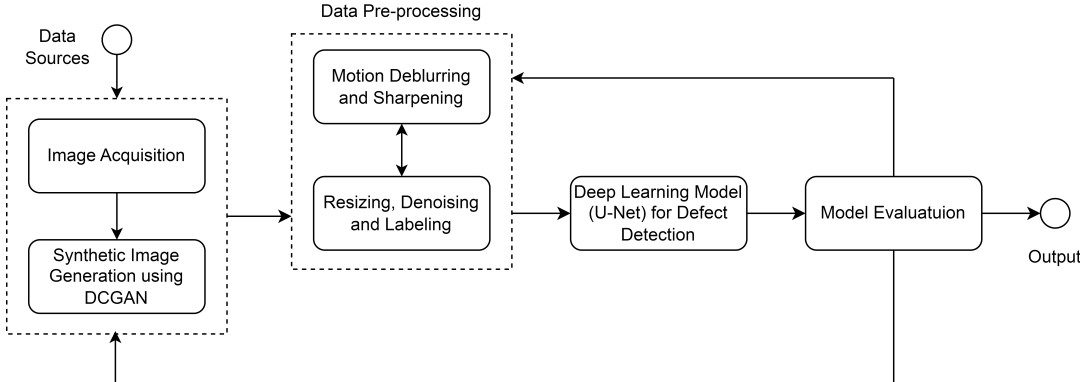

**Figure 1.** The deep learning framework for aircraft engine defect detection.

### 3.1. Image Acquisition with Synthetic Image Generation

The dataset used for this research is the stage 4 high-pressure compressor blade images of an aircraft engine, with examples shown in Figure 2. These images are created based on a borescope inspection video [20]. pytube and OpenCV are used to extract the video frames.

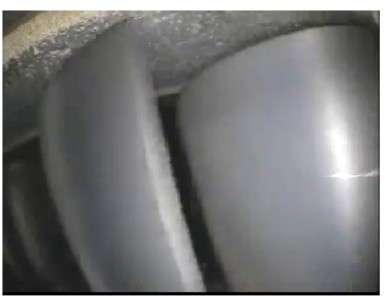 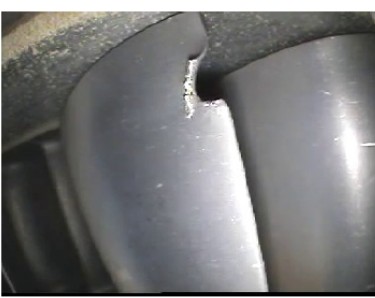 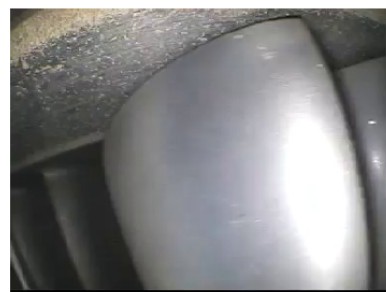

**Figure 2.** Sample images obtained through the image acquisition process.

Training deep learning models requires a large amount of data. One of the popular approaches to augment the data is to apply transformations such as rotation, flipping and zooming to images. While it enhances the deep learning model's robustness and generalisation, it does not add new information to the model. As the images in this research are obtained from a single-stage high-pressure compressor with limited defects and shapes, a model built on these images may not sufficiently generalise the data distribution. Therefore this research customises a deep convolutional generative adversarial nets (DCGAN [21]) architecture which explicitly uses convolutional and convolutional-transpose layers in the discriminator and generator respectively to generate synthetic images after the model is trained for 5000 epochs, as shown in Figure 3. As there is no standard optimisation objective available, a visual inspection of the results is implemented to determine the optimal number of training epochs in this research.

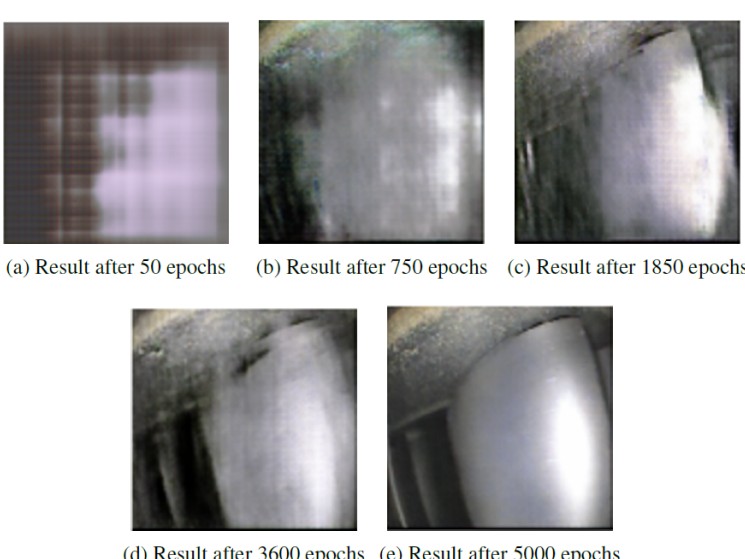

(a) Result after 50 epochs    (b) Result after 750 epochs    (c) Result after 1850 epochs

(d) Result after 3600 epochs   (e) Result after 5000 epochs

**Figure 3.** Synthetic image generation using DCGAN in 5000 epochs.

### 3.2. Data Pre-Processing

The data pre-processing module resizes, denoises and labels the images before feeding them into the deep learning model for defect segmentation. In borescope inspection, blades typically rotate as they are imaged and inspected, causing motion blur, as shown in Figure 4.

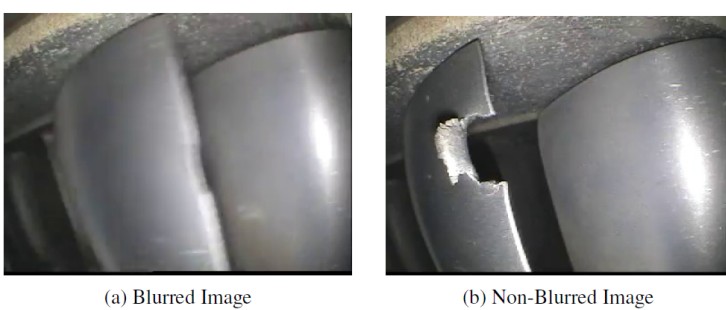

(a) Blurred Image                    (b) Non-Blurred Image

**Figure 4.** Motion blur in the input images.

For motion deblurring, the non-blind and blind motion deblurring methods are commonly used. The non-blind methods such as the Wiener filter and Richardson–Lucy algorithm require prior knowledge about the deblurring process, while the blind motion deblurring methods such as Neural networks are based on a black box approach. Since the blurring process is unknown, this research develops a hybrid model using a customised

Pix2Pix GAN [22] model in combination with computer vision methods. Pix2Pix GAN is a variant of conditional GAN that generates target images based on source images. Its generator is a U-Net-based model consisting of encoder and decoder blocks, and the discriminator is a convolutional PatchGAN classifier to identify real and fake images. The research trains the model on a series of clear and blurred images such that the model can convert between them. Images from the source and target domains are concatenated before being fed into the model. As the images produced by the generator contain noise, a denoising filter and a sharpening kernel are applied to remove the noise and sharpen the edges before feeding them into the discriminator, as shown in Figure 5.

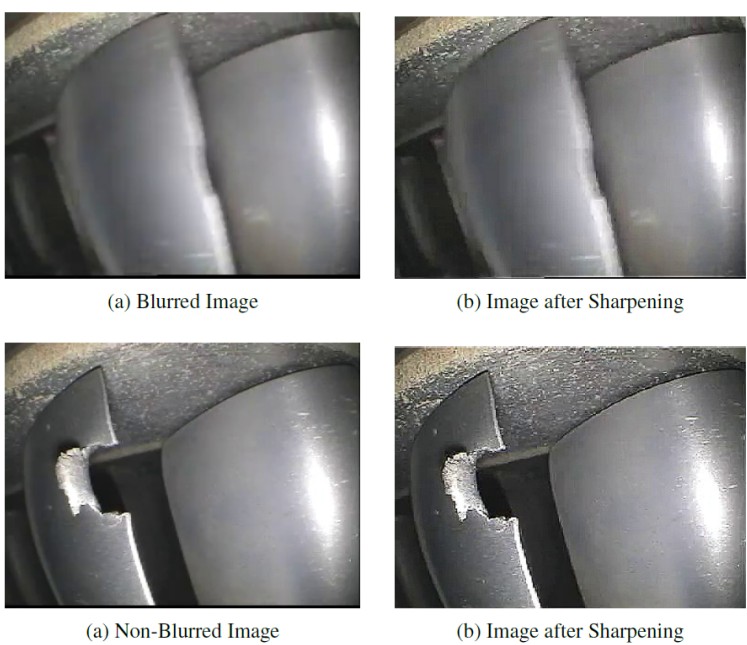

(a) Blurred Image      (b) Image after Sharpening

(a) Non-Blurred Image      (b) Image after Sharpening

**Figure 5.** Image sharpening for blurred and non-blurred images.

As shown, the denoising and sharpening process improves image quality. However, the improvement for the fuzzy image is less significant than for the clear image. This may be due to the pixel translation as the result of motion blur. To obtain an objective measure of the model performance in image deblurring, this research applies Laplacian variance to measure the image blurriness, which involves convolving images with a Laplacian filter and calculating the variances. A high variance represents the widespread edge-like and non-edge-like features in an image, showing the good quality of the image, while a low variance represents the low presence of edge-like features, which is usually the case with fuzzy images. The results in Figure 6 show that the model reduces the image blur by a factor of 10.

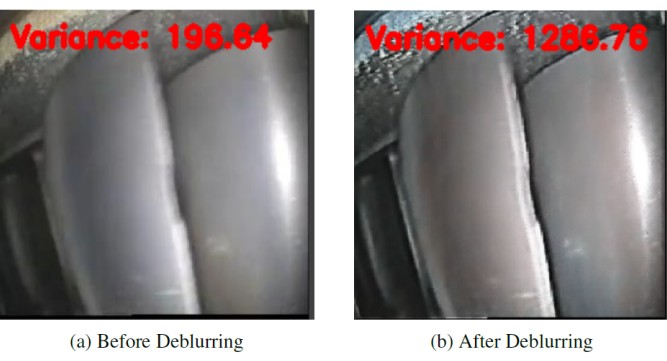

(a) Before Deblurring      (b) After Deblurring

**Figure 6.** Results after post-processing.

### 3.3. Aircraft Engine Damage Segmentation

The pre-processing also creates masks (using the VGG Image Annotator [23]) as shown in Figure 7, to identify defect areas through pixels for image segmentation. These masks are then used along with the images as input to the segmentation model.

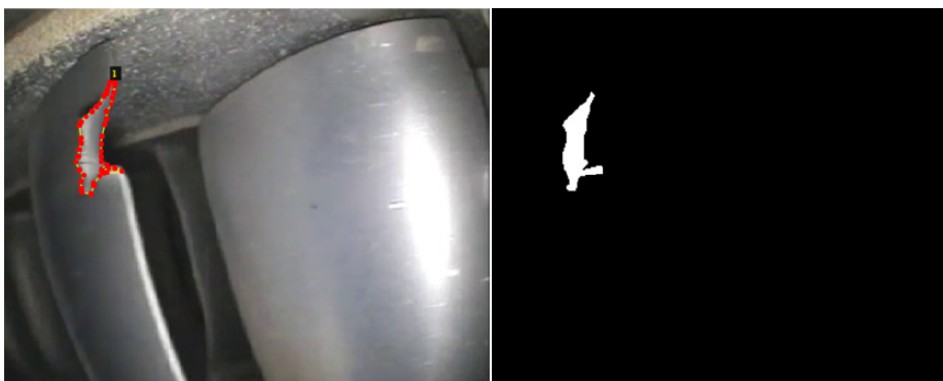

**Figure 7.** Masks for image segmentation.

Image segmentation techniques are classified into computer vision and data-driven methods. The data-driven methods are further divided into classic machine learning methods (e.g., Random Forest and SVM) and deep learning-based methods (e.g., Mask RCNN and SegNet). Although computer vision methods work well for cases where there is a clear demarcation between the target areas and the background, these methods are not suitable for aircraft engine defect images with unclear demarcation. The methods also have other weaknesses such as manual feature engineering and lack of model scalability for image sizes, which are further tested and discussed in the discussion section.

There are two common deep learning methods used to identify objects from images, i.e., bounding box or object detection approach (e.g., R-CNN and YOLO families) and instance segmentation approach (e.g., U-Net and Mask R-CNN). While the bounding box approach creates bounding boxes around the objects of interest to indicate the location and size of an object, the instance segmentation approach classifies the pixels into different instances to identify the shape of an object directly. Since the size of a defect is an important factor in determining a maintenance action (e.g., if the defect exceeds a certain threshold, emergency maintenance is required), this research adopts the instance segmentation approach to identify the accurate shape of an object. Because of the U-Net's simpler architecture and feature extraction approach over Mask R-CNN, this study customises the U-Net architecture for defect segmentation, as shown in Figure 8. It consists of an encoder block, which serves as a feature extractor for input images, and a decoder block, which combines spatial information from the encoder and decoder layer with the feature information from the encoder.

The initial U-Net is composed of two sets of three $3 \times 3$ convolution layers with Relu activation followed by a $2 \times 2$ max pooling layer for feature extraction in the contraction path. In this study it is modified to match the input image size of $240 \times 240$. The number of filters is also successively increased to extract more features. After each convolution layer, a dropout layer is added with a dropout value between 0.1 and 0.3 to generalise the model and overcome the potential overfitting. The maximum pooling layer is not included in the last set of convolution layers. The model weights are initialised from a normal distribution instead of a uniform distribution as originally presented since it is observed that initialising from a normal distribution leads to a faster training convergence. Subsequently, various loss functions are applied for model training, evaluation and result comparison.

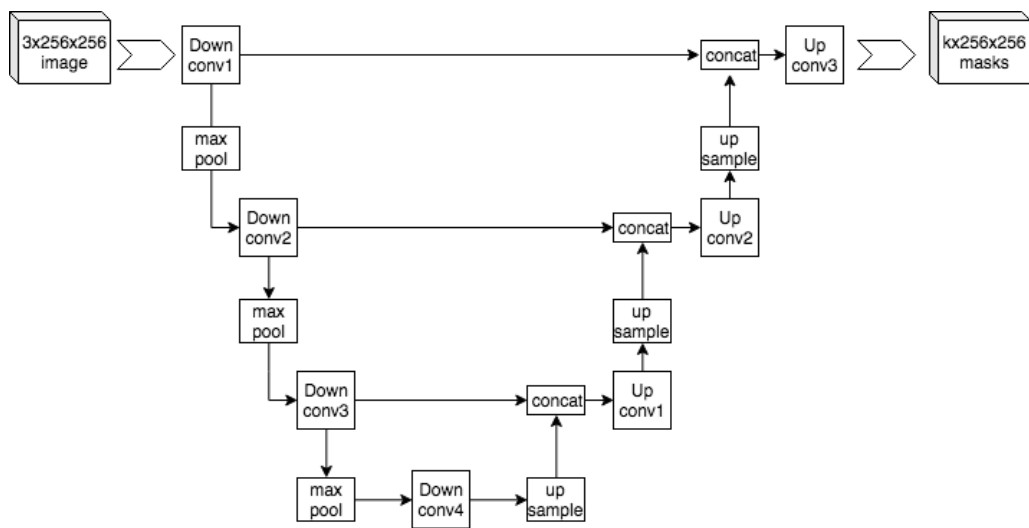

**Figure 8.** The customised U-Net architecture for aircraft engine defect segmentation.

## 4. Results

The training dataset condition (e.g., noises and small defects), model optimisation and model evaluation are tightly coupled with the loss function selection. Due to the imbalanced nature of data, restricted variation in data distribution and wide range in defect sizes, this research analyses the U-Net model results based on a confusion matrix by the loss function. Four loss functions, binary cross-entropy [24], focal loss [25], Jaccard coefficient loss [26] and Tversky loss [27], are used. Due to the limited size of training data available, the U-Net model tend to overfit when trained for more than 300 epochs across loss functions. Hence, the model performance is verified for three different epochs in the following sections, namely 50, 150 and 300 epochs.

### 4.1. Model with Binary Cross-Entropy Loss

The accuracy, precision and recall are the key indicators for model performance evaluation. The images are split into training and testing datasets at a 9:1 ratio, with a batch size of 16 to maximise the computational efficiency. The model is trained and compared at 50, 150, and 300 epochs, yielding the following results (Table 1):

**Table 1.** Results for the model with binary cross-entropy loss.

| Epochs | t_acc | val_acc | t_loss | val_loss | t_prec | val_prec | t_recall | val_recall |
|--------|-------|---------|--------|----------|--------|----------|----------|------------|
| 50     | 0.995 | 0.989   | 0.014  | 0.042    | 0.837  | 0.848    | 0.778    | 0.905      |
| 150    | 0.997 | 0.994   | 0.006  | 0.016    | 0.911  | 0.931    | 0.89     | 0.931      |
| 300    | 0.998 | 0.995   | 0.003  | 0.017    | 0.931  | 0.949    | 0.955    | 0.939      |

The results indicate that the model achieves a high level of accuracy in each epoch and the accuracy gradually increases with the number of epochs. However, since the input data are highly imbalanced, the accuracy alone should not be treated as a reliable metric of model performance. In this study, precision and recall are also used, both of which improve along with increasing epochs but exhibit no significant differences from the characteristics shown by the accuracy. Although the model is able to detect the defects of significant sizes, for very small sizes it achieves only modest success at high epochs and fails to identify them at low epochs, as shown in Figure 9 (see the Appendix A for additional information).

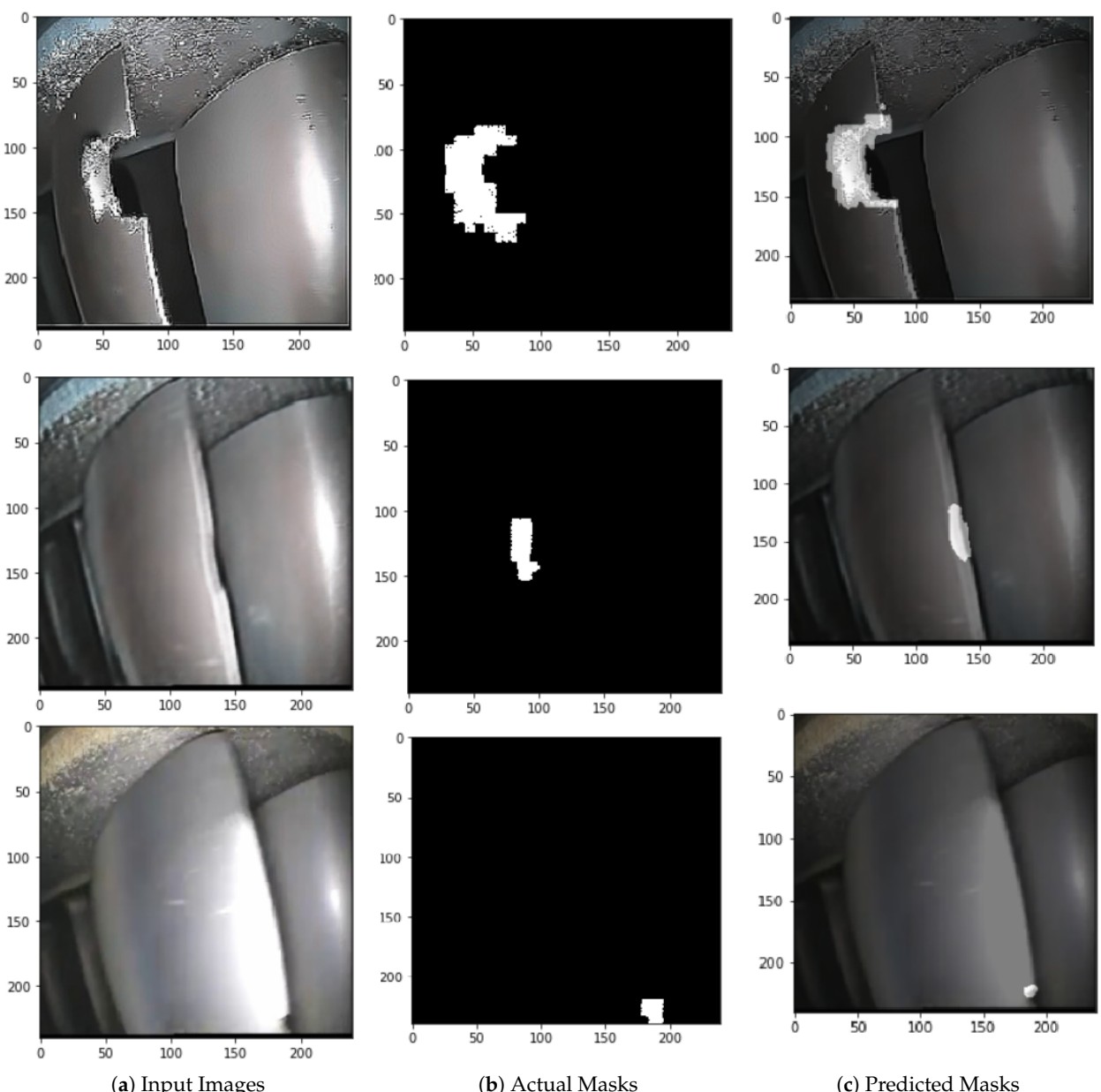

(**a**) Input Images　　　　(**b**) Actual Masks　　　　(**c**) Predicted Masks

**Figure 9.** BCE segmentation on test data after 300 epochs.

*4.2. Model with Focal Loss*

The focal loss function is useful for cases of unbalanced class distribution. The following results (Table 2) are obtained at 50, 100 and 150 epochs:

**Table 2.** Results for the model with focal loss.

| Epochs | t_acc | val_acc | t_loss | val_loss | t_prec | val_prec | t_recall | val_recall |
|--------|-------|---------|--------|----------|--------|----------|----------|------------|
| 50 | 0.99 | 0.975 | 0.001 | 0.006 | 0.953 | 0.926 | 0.205 | 0.427 |
| 150 | 0.997 | 0.994 | 0.00056675 | 0.001 | 0.962 | 0.936 | 0.817 | 0.936 |
| 300 | 0.998 | 0.995 | 0.00032251 | 0.001 | 0.976 | 0.965 | 0.887 | 0.922 |

Although the results achieve higher precisions at various epochs, the recalls are lower compared to the results from the binary cross-entropy loss model. The model still misclassifies a proportion of cases with very small defects (see the Appendix A for images).

Testing with different values of $\gamma$ could potentially lead to better recalls and ultimately improved classification for the minority classes.

### 4.3. Model with Jaccard Loss

The Jaccard loss, also known as the intersection over union (IOU) metric, is one of the commonly used loss functions for image segmentation. It measures and maximises the similarity or intersection between an actual mask and the predicted mask based on the negative Jaccard index. There is a vanishing gradient problem when the model is trained with the original design, where the loss coefficients decrease to the point that the model can no longer be reliably trained. To solve the issue, this research adds a batch normalisation layer to the model architecture following its convolution layer. The results of the model are shown in Table 3.

The model significantly underfits the data at 50 epochs with no defects predicted but improves as the epochs increase. For large defects the model predicts mask shapes well in correlation with the high Jaccard coefficient. It is observed that training with the Jaccard index provides better results compared to the previous two models. However, the model still fails to provide the correct classification for small defect cases.

**Table 3.** Results for the model with Jaccard loss.

| Epochs | t_loss | val_loss | t_coeff | val_coeff |
| --- | --- | --- | --- | --- |
| 50 | −0.3534 | −0.0158 | 0.3019 | 0.0158 |
| 150 | −0.8046 | −0.6156 | 0.7677 | 0.6156 |
| 300 | −0.8139 | −0.8892 | 0.8248 | 0.8892 |

### 4.4. Model with Tversky Loss

The Tversky index is a measure of similarity and a generalisation of the Jaccard index and dice coefficient. The loss function used with the Tversky index is called the focal Tversky loss. By setting its coefficients $\alpha \geq \beta$ and adjusting $\gamma$ accordingly, the false negative issue encountered in previous models can be penalised, thus ensuring that the minority classes such as small defects are classified correctly. The model results are shown in Table 4.

**Table 4.** Results for the model with Tversky loss.

| Epochs | t_loss | val_loss | t_coeff | val_coeff |
| --- | --- | --- | --- | --- |
| 50 | 0.1844 | 0.525 | 0.896 | 0.576 |
| 150 | 0.2023 | 0.1178 | 0.8847 | 0.942 |
| 300 | 0.1766 | 0.1146 | 0.9021 | 0.944 |

Similar to the Jaccard loss model, the focal Tversky loss model does not converge at 50 epochs, as evidenced in Figure 10.

The model is able to detect some large defects and the boundary pixels are also classified well in some cases. It is subsequently trained up to 150 epochs and above until it converges and stabilises at 300 epochs as shown in Figure 11.

At 300 epochs, the model is able to predict the masks for cases where the defect areas are defined well, although there are a few misclassifications at the boundary. The model also well predict cases with no defect. However, it still fails to predict some very small defect cases, as shown in Figure 12.

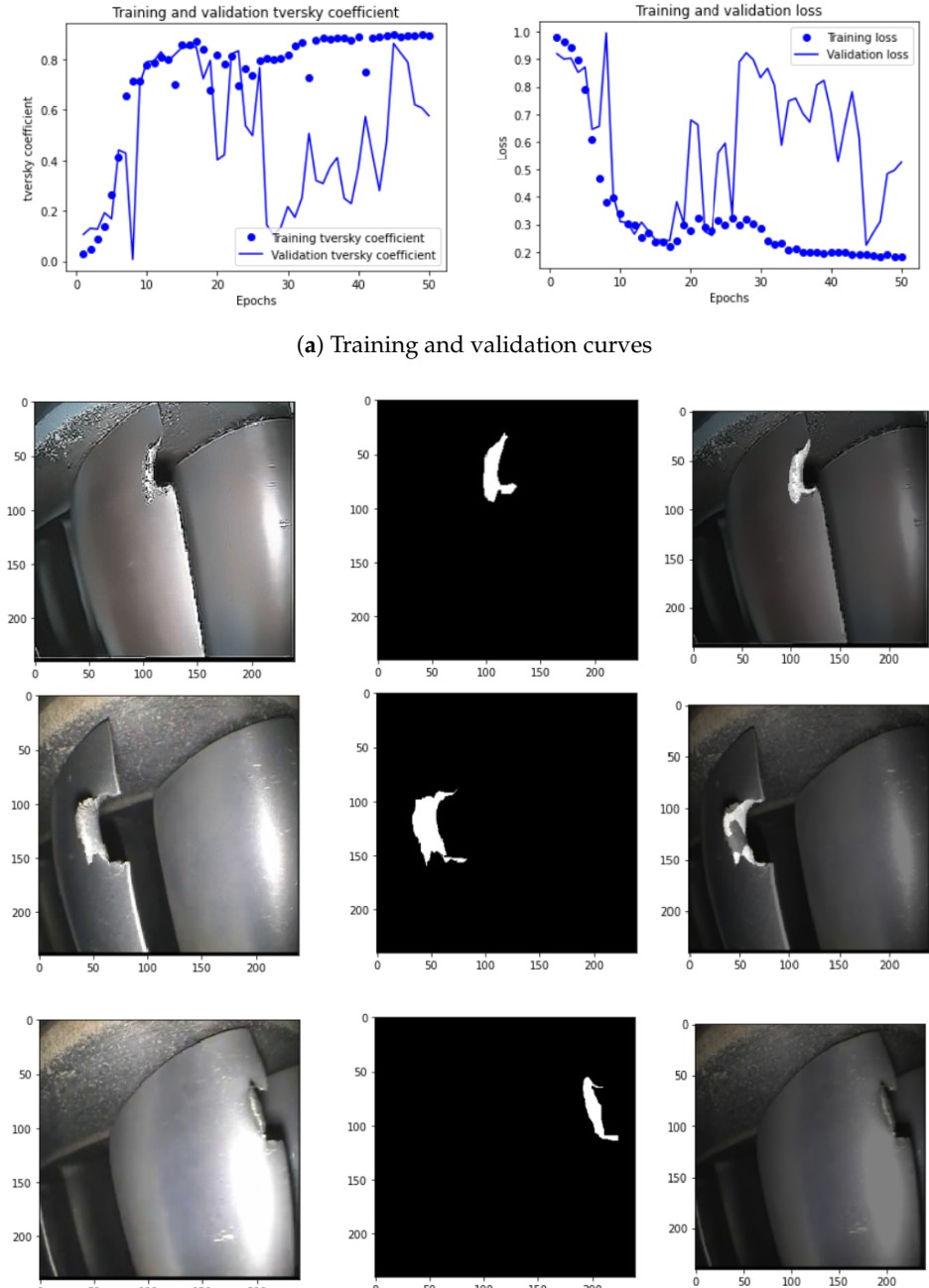

(**a**) Training and validation curves

(**b**) Input images, and corresponding actual and predicted masks

**Figure 10.** Results for the model with focal Tversky loss at 50 epochs.

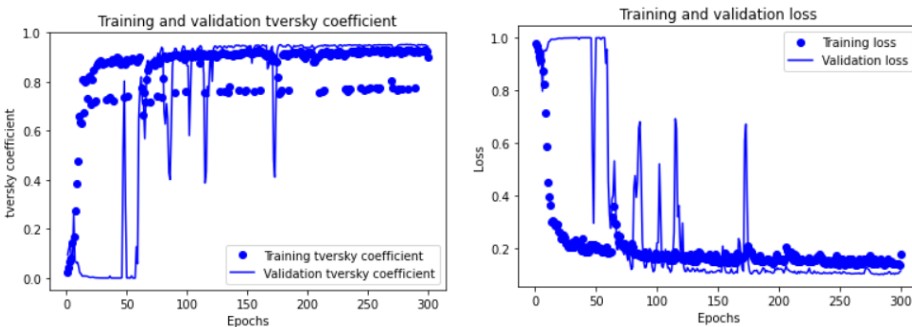

**Figure 11.** TL training and validation curves after 300 epochs.

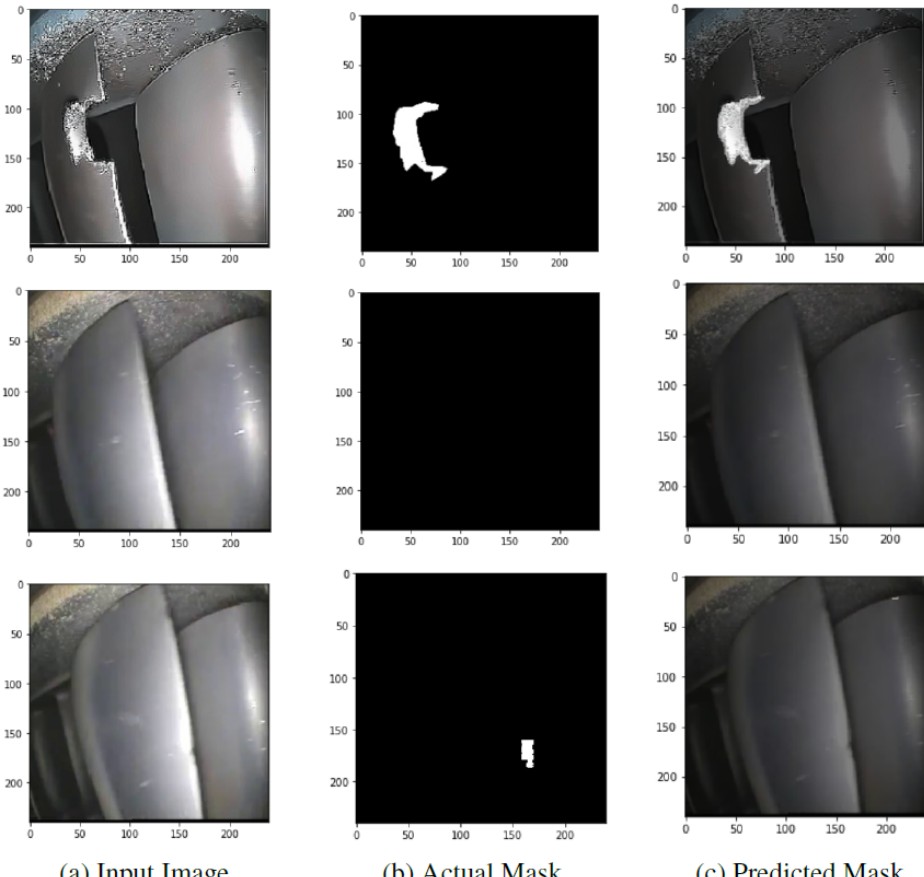

(a) Input Image               (b) Actual Mask             (c) Predicted Mask

**Figure 12.** TL segmentation on test data after 300 epochs.

## 5. Discussion

The deep learning framework developed for automated borescope inspection provides dual benefits, namely reducing the time required to perform otherwise manual aircraft engine borescope inspection and improving the overall safety of the aircraft free from human error. The framework includes an image processing component for motion deblurring and sharpening based on the integration of classic computer vision algorithms and Pix2Pix GAN, an adjusted GAN model for synthetic image generation and a customised U-Net for engine defect detection. This research applies an empirical and comparative approach by implementing multiple algorithms in each component, specifically image capturing and GAN model for image acquisition, image denoising, motion deblurring and image sharpening for data pre-processing, and a customised U-Net with various loss functions for defect segmentation.

Current approaches to image deblurring are either kernel-based or deep convolutional-based methods. This research develops a hybrid model by integrating classic computer vision methods and deep learning. After sharpening the images, an initial experiment is also conducted to study whether image blurring has an impact on image segmentation, with the results shown in Figure 13.

The above results show that the deep learning model cannot locate the defect areas of fuzzy images. Since this is very common in aircraft engine images, it may be necessary to pre-process the fuzzy images.

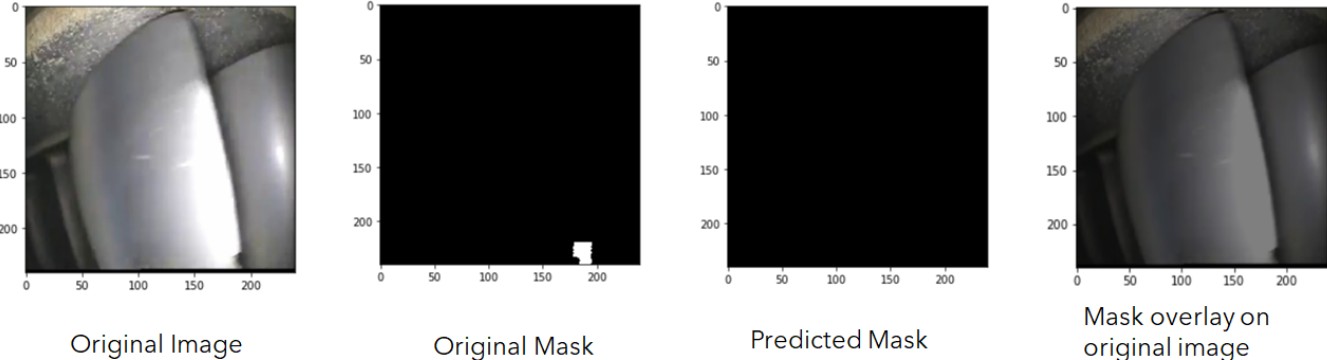

Original Image  Original Mask  Predicted Mask  Mask overlay on original image

**Figure 13.** Image segmentation using blurred images.

While a $10\times$ improvement in image quality is observed from the hybrid model, there are some weaknesses of the model. The Pix2Pix GAN model occasionally suffers from the convergence problem, where the model cannot find an equilibrium between the discriminator and generator when the discriminator loss reaches zero and does not change for a number of iterations. In addition, the filters used in denoising and sharpening are fixed and do not adjust, and sometimes introduce new noises to the images.

Since a deep learning-based instance segmentation model is computationally expensive, this research also experiments with a computer vision-based image segmentation approach based on thresholding for binary classification. The approach classifies the pixels of an image into two groups according to their intensity (i.e., 0 or 1 corresponding to a threshold). In particular, this research uses Otsu thresholding to minimise the within-class variance and then the Otsu filter to segment the images. However, this method does not work for the cases where the pixel intensities are similar between the defect areas and the image background (see images in the Appendix A).

If the data are insufficient to train deep learning models, classic computer vision-based approaches such as the Random Forest classifier tend to offer better solutions. These approaches often classify the pixels of an image into different classes based on the features obtained from the pixel values, therefore segmenting the objects from the background. This research develops a Random Forest classifier and generates a total of twenty-nine features with binary masks for the classifier using the Gabor filter, Canny edge detector filter, Roberts edge detector filter, Sobel filter, Scharr filter, Prewitt filter, Gaussian filter with sigma value 3, Gaussian filter with sigma value 7, median filter with sigma value 3 and variance filter with size 3 (see detail in the Appendix A). The Random Forest classifier achieves an average precision of 0.76 with the testing dataset compared to over 90% from the deep learning models. Although it performs well on new images with similar data distributions as the training dataset, it fails to make accurate predictions for images with significantly different distributions, as shown in Figure 14.

The Random Forest classifier is not scalable. Using each pixel as a training data point results in a large and unbalanced dataset with even few images. In addition, the feature extraction is done manually on a trial-and-error basis, and there is no one-size-fits-all filter found to get the most informative features, which can be time-consuming. To automate the feature engineering and efficient modelling, the research also investigates the transfer learning approach using a VGG16 model pre-trained on an ImageNet dataset. The VGG16 model is a type of convolutional neural network for image classification, consisting of convolutional layers followed by Relu activation, max pooling layers and a softmax output layer. To keep the input shape in congruence with the label shape, only the first two layers of the model are used for feature extraction in this study. The input layer is adjusted to fit the image size (i.e, $256 \times 256 \times 3$). The features are transformed into single columns and the labels are obtained by applying image masks. On this basis, the Random Forest model is retrained and tested. However, the model achieves a lower average precision of 0.69 (see

model architecture and results in the Appendix A) than the manual feature engineering approach above. This may be due to a highly imbalanced dataset, different data distribution with the ImageNet dataset and inefficient auto-extracted features. These two methods and their results can be used as benchmarks for developing new image segmentation models using various techniques and datasets.

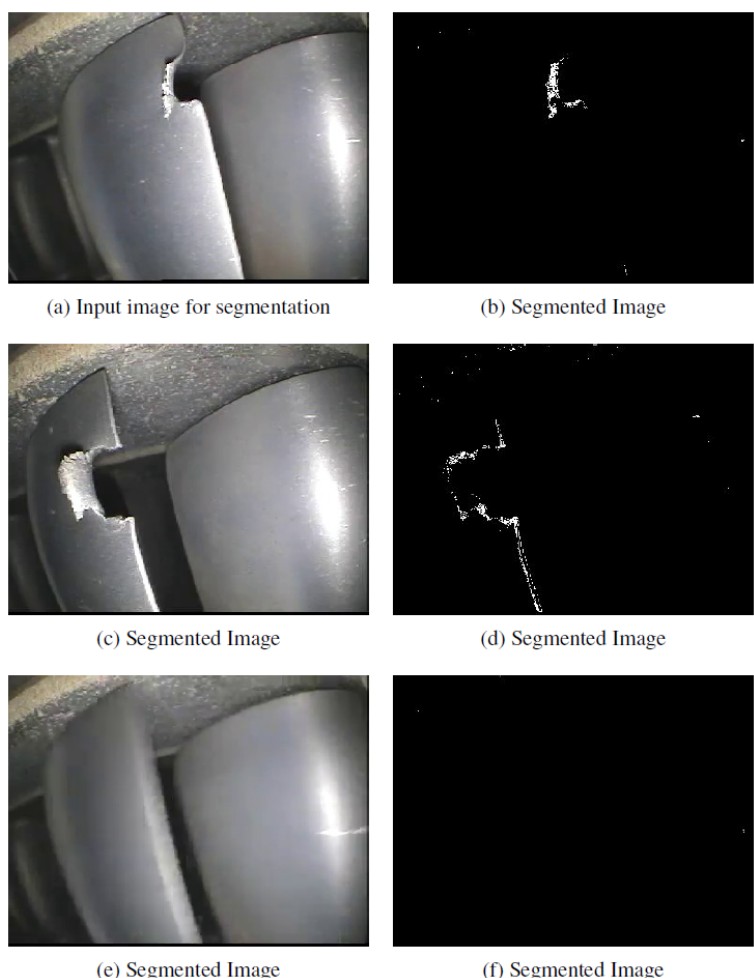

(a) Input image for segmentation        (b) Segmented Image

(c) Segmented Image        (d) Segmented Image

(e) Segmented Image        (f) Segmented Image

**Figure 14.** Image segmentation using Random Forest.

This research customises a U-Net for defect segmentation. Small defects in the images cause a class imbalance problem, so it is necessary to try different loss functions. The model is successful in detecting defects in images that have undergone data pre-processing and deblurring, and achieves the precisions and recalls of over 90% with these loss functions. For the balanced datasets, the binary cross-entropy loss trained with high epochs performs well, though there are still some boundary layer misclassification. The focal loss and Tversky loss are effective for unbalanced datasets, producing good results with fewer epochs than other loss functions, and no boundary layer misclassifications are observed. It is difficult to train the model with Jaccard loss. Although it can converge eventually, the fluctuation between training and validation indicates that its training is unstable. In addition, it does not perform well on imbalanced datasets. It is evident that the choice of loss function affects the efficiency of model training and the quality of results. Despite the positive findings from focal loss and Tversky loss, overall the model is unable to identify very small defects. Different loss coefficient values need to be further explored.

Given the limited experimental cases and lack of variety of image distributions used in the research, further validation of the proposed framework based on an enhanced image acquisition module is necessary. Moreover, the GAN models for synthetic data generation

are prone to so-called mode collapse, where the generator produces an image essentially mapping every random input vector. This research tests solutions to solve the problem empirically by adding random Gaussian noises to the generator and batch normalisation layers to the generator and discriminator, switching the loss function from binary cross-entropy to Wasserstein loss, and fuzzfying the zero and one hard labels. However, the problem remains. This may also be due to a lack of variation in the distribution of training data. Further study of larger datasets with different types of defect shapes is required.

This research applies an empirical and comparative approach by implementing multiple algorithms and diverse parameters in each component of the proposed framework. It may be necessary to establish a model benchmark for engine defect detection on the basis of the existing literature and compare the results with the benchmark. With further ablation studies to verify the effectiveness, each component of the proposed framework may well serve as an independent application in related industries.

## 6. Conclusions

This research develops a novel framework for automated borescope inspection of aircraft engines based on deep learning and computer vision. In the framework, a customised U-Net architecture is included to detect the defects on high-pressure compressor blades. The framework also includes an image processing component for motion deblurring and sharpening and an adjusted GAN model for synthetic image generation. This research applies an empirical and comparative approach by implementing multiple algorithms in each component. The framework achieves a precision and recall of over 90%. The hybrid model for motion deblurring in the framework results in a $10\times$ improvement in image quality.

This research trains the customised U-Net model with four loss functions for data imbalance and small size of defects problems, as well as for model results comparison and evaluation. The model is successful in detecting defects in images that have undergone data pre-processing and deblurring. The focal loss and Tversky loss give good results with fewer epochs than other loss functions without boundary layer misclassifications and are effective for unbalanced datasets. For very small sizes of defects, the framework only achieves modest success with cross-entropy loss and focal loss. Although the framework focuses on defect detection for aircraft engines, it can be customised for other industries with further adaptation and testing. Future work includes the generalisation of the deep learning framework, larger datasets and the loss function parameter grid search to solve the data imbalance and small defect detection problems.

**Author Contributions:** Conceptualization, A.U., J.L., S.K. and S.A.; methodology, A.U. and J.L.; software, A.U.; validation, A.U. and J.L.; formal analysis, A.U.; investigation, A.U.; resources, A.U.; data curation, A.U.; writing—original draft preparation, A.U.; writing—review and editing, A.U., J.L. and S.A.; visualization, A.U.; supervision, J.L., S.K. and S.A.; project administration, J.L. All authors have read and agreed to the published version of the manuscript.

**Funding:** This research was funded by Innovate UK [Grant No. 103082-263289; Project No. 113174; Project title. Digitally Optimised Through-life Engineering Services—(TES Digital)].

**Data Availability Statement:** Publicly available datasets were analysed in this study. The data is openly available from [RB211 535 HPC4, RVI Ltd. Remote Visual Inspections, YouTube.com] at [https://www.youtube.com/watch?v=eO6GRU4RfC4].

**Conflicts of Interest:** The authors declare no conflict of interest.

## Appendix A

*Appendix A.1. Results*

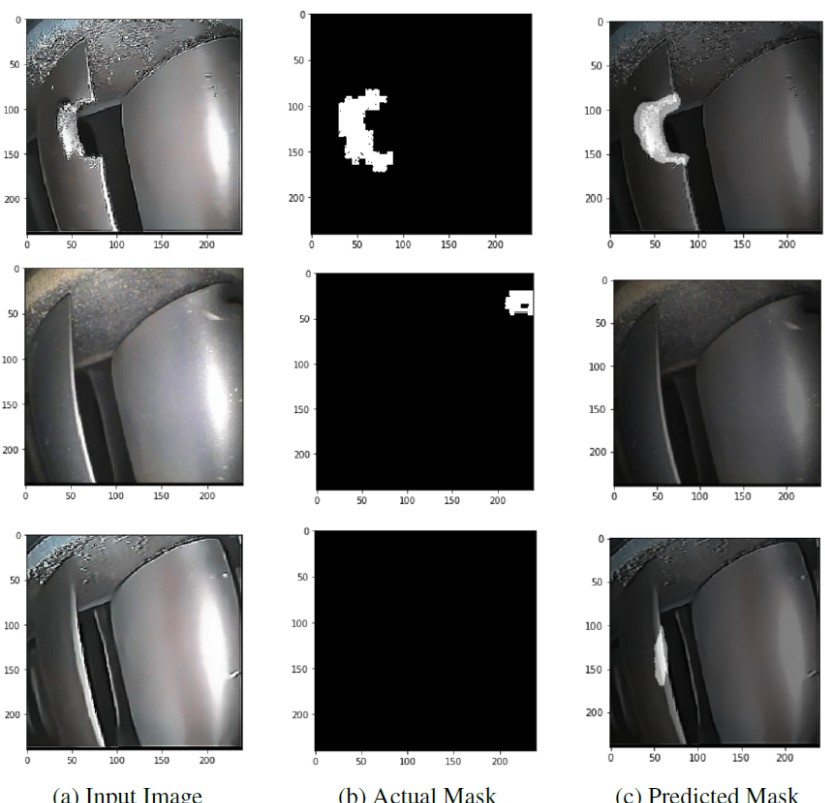

(a) Input Image                (b) Actual Mask                (c) Predicted Mask

**Figure A1.** BCE segmentation on test data after 50 epochs.

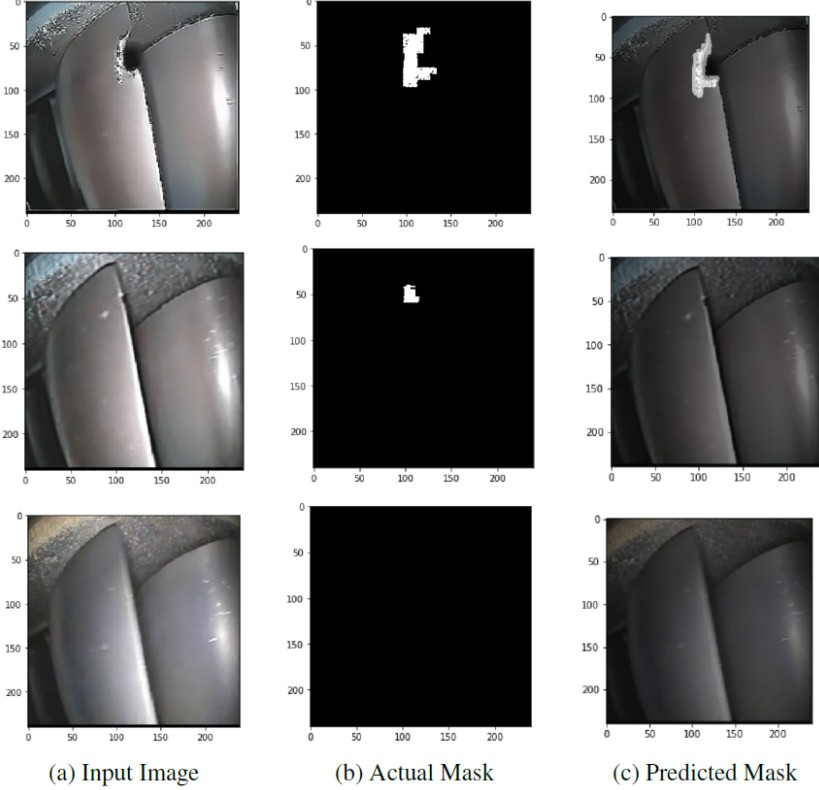

(a) Input Image                (b) Actual Mask                (c) Predicted Mask

**Figure A2.** BCE segmentation on test data after 150 epochs.

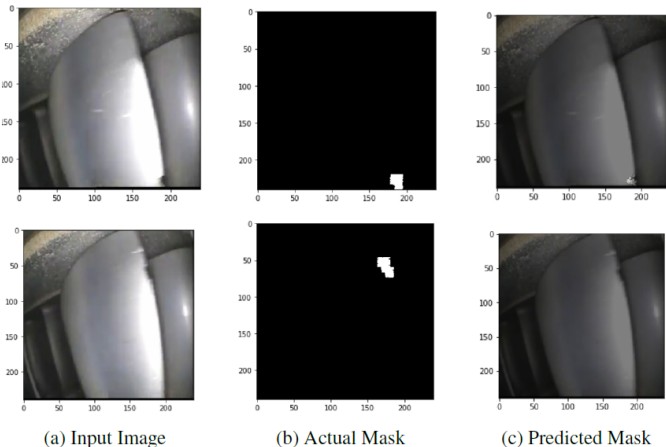

(a) Input Image      (b) Actual Mask      (c) Predicted Mask

**Figure A3.** FL segmentation on test data after 50 epochs.

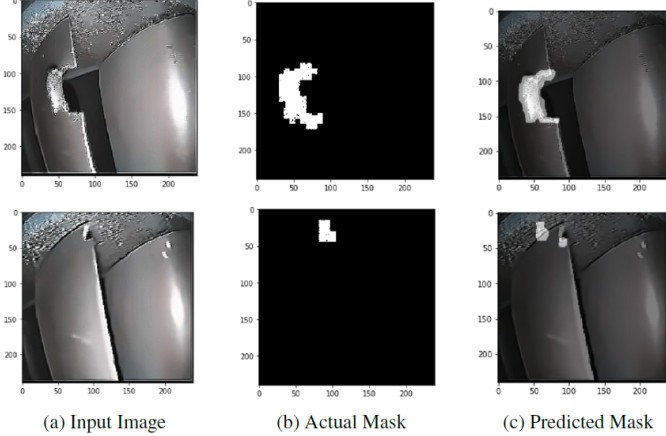

(a) Input Image      (b) Actual Mask      (c) Predicted Mask

**Figure A4.** FL segmentation on test data after 150 epochs.

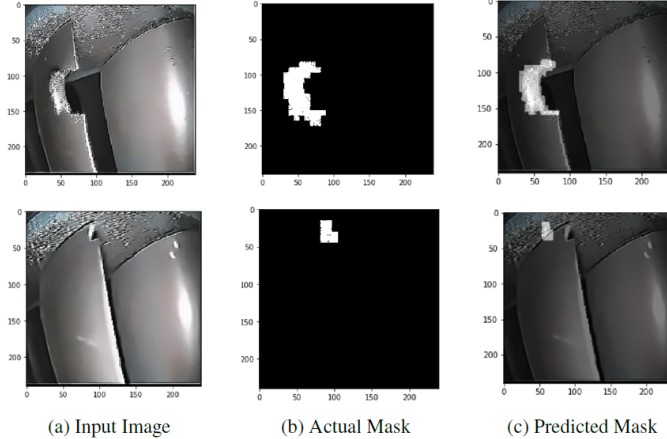

(a) Input Image      (b) Actual Mask      (c) Predicted Mask

**Figure A5.** FL segmentation on test data after 300 epochs.

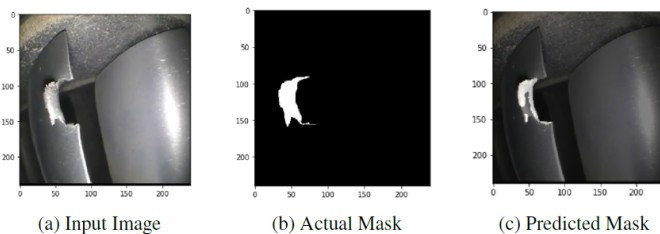

(a) Input Image      (b) Actual Mask      (c) Predicted Mask

**Figure A6.** JL segmentation on test data after 150 epochs.

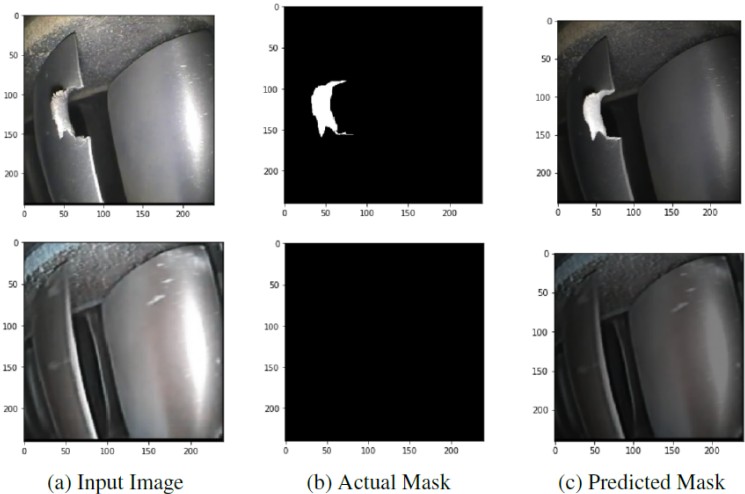

(a) Input Image        (b) Actual Mask        (c) Predicted Mask

**Figure A7.** JL segmentation on test data after 300 epochs.

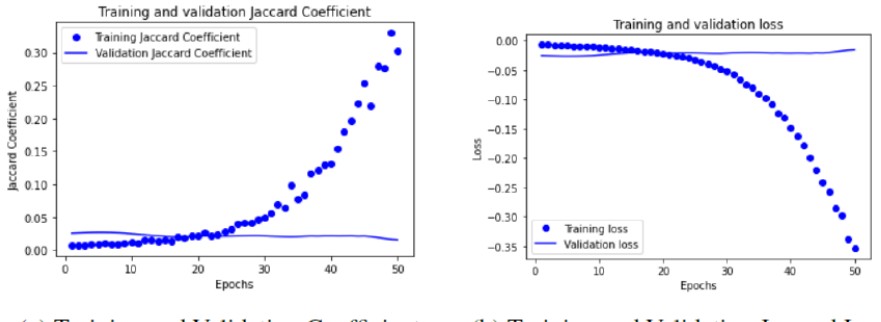

(a) Training and Validation Coefficient    (b) Training and Validation Jaccard Loss

**Figure A8.** JL training and validation curves after 50 epochs.

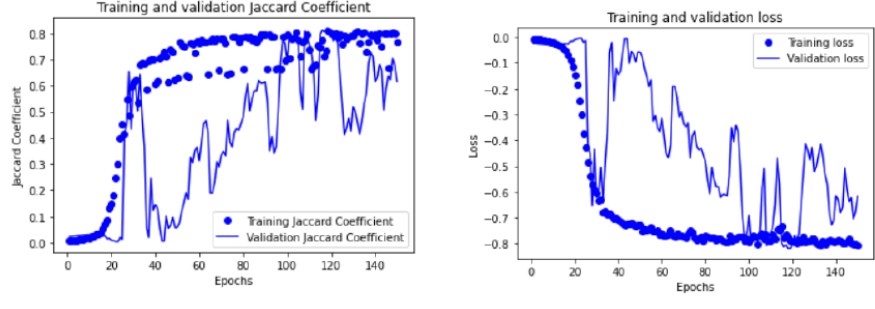

(a) Training and Validation Coefficient    (b) Training and Validation Jaccard Loss

**Figure A9.** JL training and validation curves after 150 epochs.

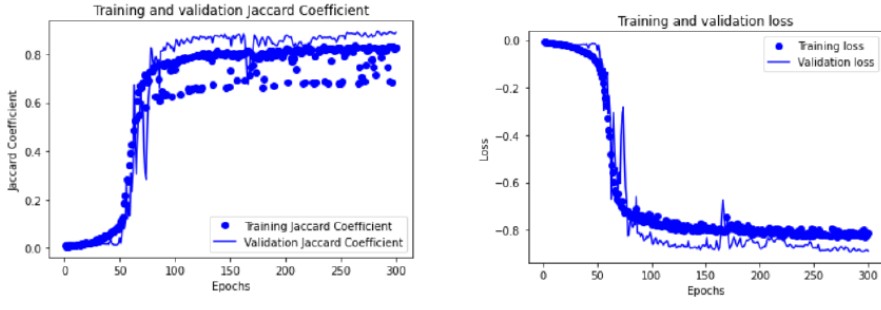

(a) Training and Validation Coefficient    (b) Training and Validation Jaccard Loss

**Figure A10.** JL training and validation curves after 300 epochs.

*Appendix A.2. Discussion*

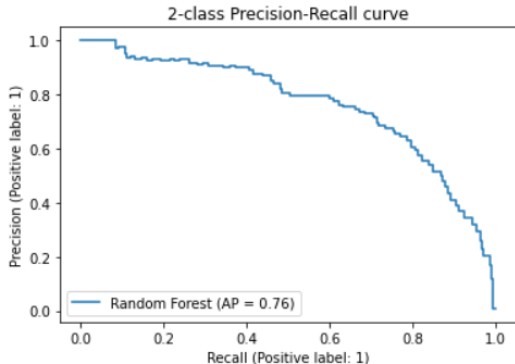

**Figure A11.** Precision–recall curve for the Random Forest model based on feature engineering.

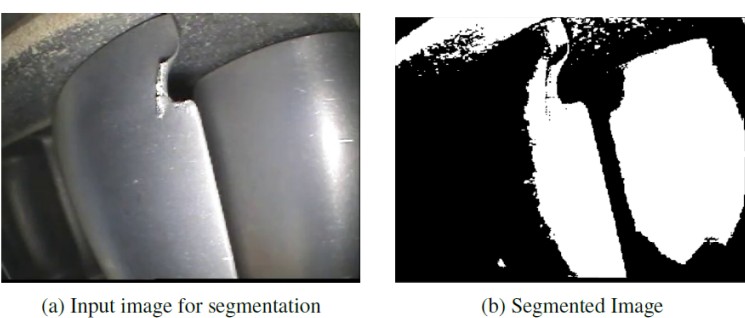

(a) Input image for segmentation  (b) Segmented Image

**Figure A12.** Image segmentation using the Otsu thresholding method.

```
Gaussian s7        8.908375e-02
Prewitt            7.189091e-02
Sobel              6.597737e-02
Scharr             6.298847e-02
Gabor32            5.712093e-02
Roberts            5.461711e-02
Gaussian s3        5.272231e-02
Median s3          5.109089e-02
Original Image     4.982534e-02
Gabor7             4.977730e-02
Gabor31            4.870391e-02
Gabor21            4.555247e-02
Gabor8             4.293320e-02
Gabor6             4.216324e-02
Gabor29            4.166730e-02
Variance s3        4.097081e-02
Gabor30            4.072857e-02
Gabor24            3.690253e-02
Gabor5             1.851628e-02
Gabor23            1.415688e-02
Canny Edge         1.346770e-02
Gabor22            6.666496e-03
Gabor4             1.256729e-03
Gabor12            5.963830e-04
Gabor11            4.293953e-04
Gabor3             1.508139e-04
Gabor20            3.354006e-05
Gabor28            9.353051e-06
Gabor27            3.159329e-08
```

**Figure A13.** Key features evaluated from the trained Random Forest classifier, where the mean and standard deviation of the accumulation of the impurity decrease inside each tree are computed to determine the scores.

```
Layer (type)                  Output Shape                Param #
=================================================================
 input_1 (InputLayer)         [(None, 256, 256, 3)]       0

 block1_conv1 (Conv2D)        (None, 256, 256, 64)        1792

 block1_conv2 (Conv2D)        (None, 256, 256, 64)        36928

=================================================================
Total params: 38,720
Trainable params: 0
Non-trainable params: 38,720
_________________________________________________________________
```

**Figure A14.** Feature extractor model structure for transfer learning.

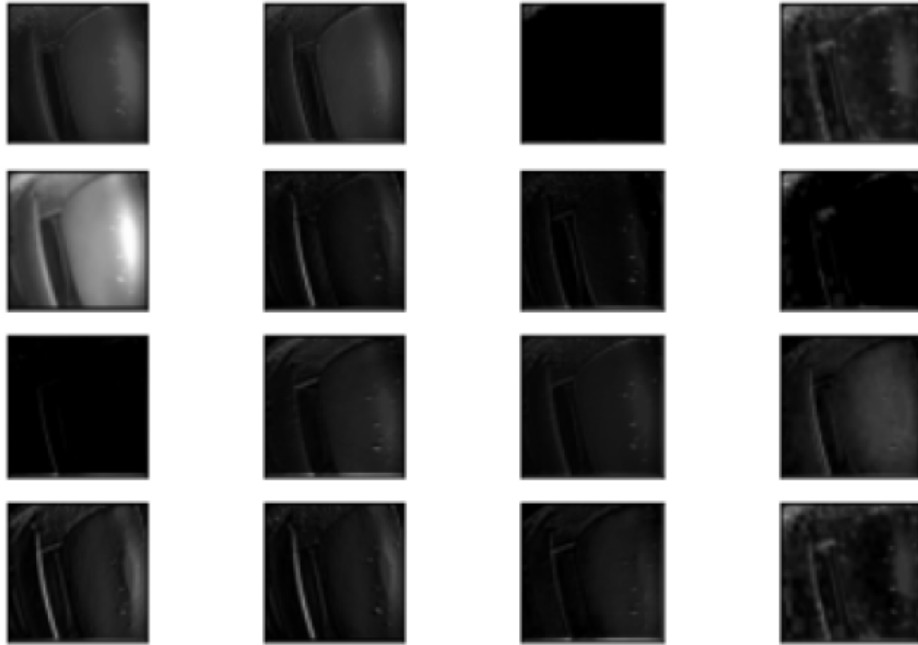

**Figure A15.** Extracted feature samples using the VGG16 model.

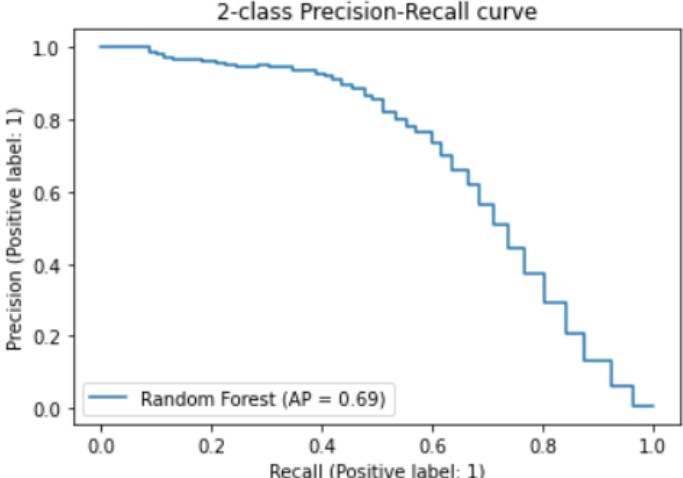

**Figure A16.** Precision–recall curve for the Random Forest model based on transfer learning.

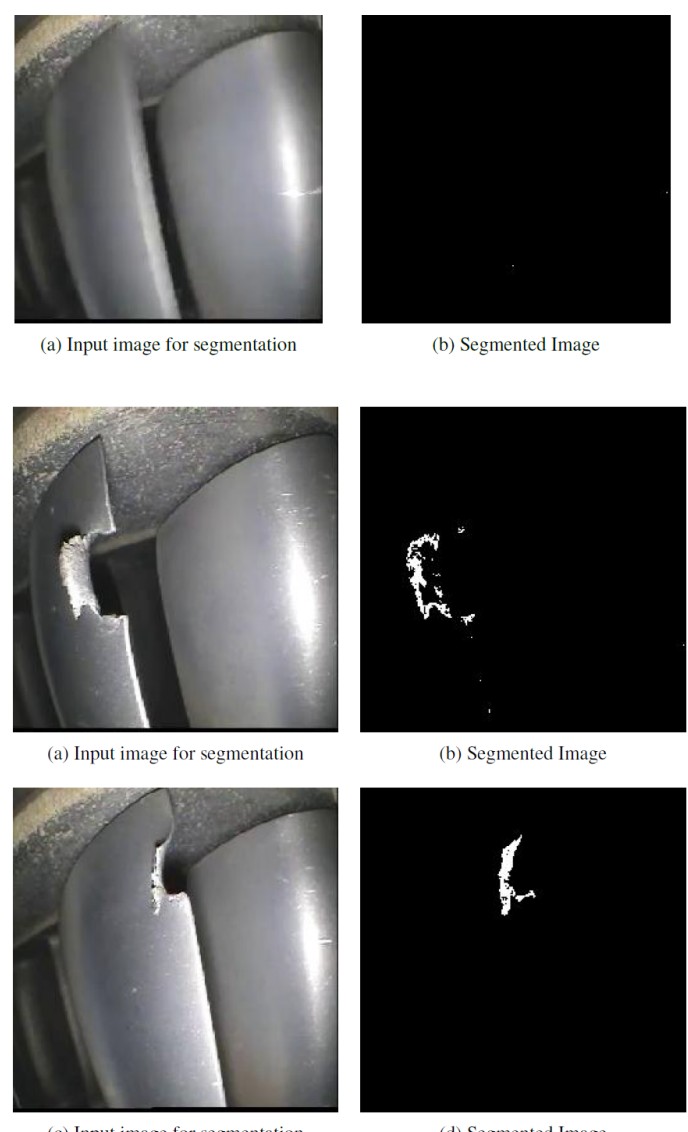

(a) Input image for segmentation

(b) Segmented Image

(a) Input image for segmentation

(b) Segmented Image

(c) Input image for segmentation

(d) Segmented Image

**Figure A17.** Random Forest segmentation on new images based on transfer learning.

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
