# Peer review of "A Deep-Learning-Based Approach for Aircraft Engine Defect Detection"

_machines, doi:10.3390/machines11020192_

Round 1

Reviewer 1 Report

1. The ambiguity of the title and problem definition: The reviewer thinks the term ‘AI Based Approach’ in the title is too broad. The reviewer recommends the authors specify the title with the proposed method. Also, the reviewer thinks the problems are not well-defined. Defining multiple problems (e.g., defect detection, data imbalance, and synthetic data generation) as one unified problem is recommended.

2) The lack of case studies: The reviewer thinks the results of experimental studies are insufficient to assert the validity of the proposed method. The reviewer suggests that the authors add an ablation study to verify the effectiveness of each module, such as synthetic image generation module and motion deblurring module. Also, the reviewer suggests that the authors add other existing algorithms, such as the hybrid method in Ref. [4], for comparison between the proposed and existing methods.

3) The lack of a logical illustration: The reviewer thinks the flowchart in Figure 1 is inappropriate for demonstrating the proposed method. In Section 3.1, the terms ‘Image Acquisition’ and ‘Synthetic Data Generation’ are described as a single module. In Section 3.2, the terms ‘Data Pre-processing’ and ‘Motion Deblurring and Sharpening’ are described as a single module. However, the description is not well aligned with the flowchart. Also, a breakpoint is not included in the feedback loop of the flowchart. The reviewer suggests that the authors modify the flowchart in terms of alignment and content.

The reviewer thinks loss function comparison is irrelevant to the research topic in terms of the defined problem or proposed method. The reviewer recommends explaining the proper reason why the various loss functions are used in the experimental studies.

The reviewer thinks redundant contents are repeated in different sections, such as comments on Ref. [1-3] in the Introduction, the Literature Review, and the model description in the first paragraph of Discussion and Conclusion.

Reviewer 2 Report

This research developed an AI-based automated borescope inspection framework using a bespoke U-Net architecture to detect defects on high-pressure compressor blades with different loss functions to address data imbalance and small defect size. A GAN model was used to generate synthetic data, a motion deblurring method was applied to improve image quality. The results showed precisions and recalls over 90%, but more research is needed for small defect detection and general borescope inspection. The work in this manuscript is interesting and meaningful. This manuscript can be considered for publication provided that the authors make careful revision following the comments listed below:

1. It is suggested that sections 1 and 2 be combined into a single section involving background, literature review and the brief introduction of this work.

2. The main contributions of the work should be highlighted in the manuscript. It is essential to specify the key point of the method, whether it is the use of different loss functions or data generation techniques.

3. Figure 1 should be more informative and provide details on the DCGAN and Unet module used in the research.

4. The text should clearly indicate the figures used to support the claims made in the manuscript.

5. For the fault detection and condition monitoring of rotating systems, beside the development of artificial intelligent techniques and other signal processing approaches, recently more and more studies on smart sensing systems based on energy harvesting techniques, such as Energy, 2022, 238, 121770; Smart Materials and Structures, 2022, 31(12), 125008; Nano Energy, 2020, 75, 104853. It is highly suggested that brief discussion on the effects of hardware development on rotor fault detection and its relation between signal processing techniques be added in Introduction.

6. It is suggested that the optimization objective should be clearly defined and supported by appropriate formulaic references.

7. The manuscript should explain what makes the method unique. It utilizes multiple image processing components, deep learning models such as U-Net and GAN, and various loss functions, but it is essential to highlight how these elements work together to differentiate the method from others.

Round 2

Reviewer 2 Report

This revised manuscript can be accpeted for publication. Thanks.